# Wage Penalties or Wage Premiums? A Socioeconomic Analysis of Gender Disparity in Obesity in Urban China

**DOI:** 10.3390/ijerph17197004

**Published:** 2020-09-24

**Authors:** Jiangli Dou, Limin Du, Ken Wang, Hailin Sun, Chenggang Zhang

**Affiliations:** 1School of Economics, Zhejiang Gongshang University, Hangzhou 310012, China; jianglidou@zjgsu.edu.cn; 2School of Economics, Zhejiang University, Hangzhou 310027, China; dlmzju@zju.edu.cn; 3School of Social Sciences, Tsinghua University, Beijing 100084, China; hjwk8586@163.com; 4The Institute for Social Governance and Development, Tsinghua University, Beijing 100084, China; 5Department of Sociology, Tsinghua University, Beijing 100084, China

**Keywords:** obesity, overweight, income, gender

## Abstract

Global obesity as a major public health problem has increased at pandemic rate, with men often outpacing women. Survey data show that the overall prevalence of obesity is higher among women than men, yet in high-income developed countries, the prevalence of overweight is higher among men than women. The differential impact of different economic stages has prompted research in transition economies such as China. Using an instrumental variable approach based on a sample of 13,574 individuals from nine provinces in the Chinese Household Income Project (CHIP), we find a 7% excess-weight premium in wages for overweight men and a 4.6% penalty for overweight women, compared to their healthy-weight peers. We also find an inverse u-shaped association between the body mass index (BMI) and logarithm of monthly income for men, with an implied optimum above the threshold of obesity, while women are better off the slimmer they are. The excess-weight premium in wages for Chinese urban men might be associated with entrenched business practices of excessive dining and drinking associated with senior positions. Policies aimed at reducing obesity in China must be adapted to its unique sociocultural context in order to have gender-differentiated effects.

## 1. Introduction

Global obesity as a major public health problem has increased at pandemic rate, with age-standardized prevalence of obesity more than tripling in men (from 3.2% to 10.8%) and doubling in women (from 6.4% to 14.9%) between 1975 and 2014 [1]. In China, the world’s most populous country, the rate of increase was even higher. In the decade 2004-2014 alone, the prevalence of obesity in China more than tripled, reaching an overall prevalence of obesity at 14% for men and 14.1% for women [2].

Global survey data show that the overall prevalence of obesity is higher among women than men, yet in high-income developed countries, the prevalence of overweight is higher among men than women [3,4,5]. Current knowledge suggests that different socioeconomic dynamics around the world contribute to the complexity of gender disparities in excess weight gain. In developed countries, where most occupational roles have been sedentary for the past half century with little gender difference, dietary intake rather than physical activities is responsible for gender disparities in overweight and obesity. However, in developing countries, the transition from agricultural labor to wage labor has led to a greater reduction in women’s physical activity than men’s, and the reduction has been particularly pronounced in rural areas. With the dramatic changes in the economic landscape, China’s obesity patterns increasingly resemble that of developed countries [6,7,8,9]. But not much is known about the extent to which excess weight gain exacerbate gender disparities in China’s socioeconomic dynamics.

Obesity is not only a major risk factor for all-cause mortality and a number of non communicable diseases such as high blood pressure, type 2 diabetes, heart disease [10,11,12,13,14,15,16,17,18,19], but also an independent risk factor for death in Coronavirus Disease 2019 (COVID-19) patients, especially in men [20]. In addition, obesity often translates into inequalities in the employment settings, making obese individuals vulnerable to social injustice [21,22,23,24]. Thus, the rapid growth of overweight and obesity is not only a health issue, but also a socioeconomic one. As such, it has stimulated the interest of researchers to investigate its causes and consequences. Lower income has been repeatedly found to be associated with a higher risk of obesity, though this relationship can be interpreted in two directions: the causal hypothesis explaining income as a cause of obesity [25,26] and the reverse causality of obesity as a cause of subsequent lower income [27,28].

Much of the literature has found that people who are overweight or obese are less likely to be employed and tend to earn lower incomes [29,30,31,32,33,34,35,36,37,38,39,40,41,42,43,44,45,46,47,48,49]. Overweight and obese people drift into lower-paying jobs primarily through two channels: reduced productivity due to poor health [50,51,52,53,54,55,56,57], and labor market discrimination that associates weight with personality traits or physical attractiveness [58,59,60,61]. However, the results are not conclusive, especially with regard to gender disparities. For example, Lundborg et al. [42] found that the probability of being employed is significantly lower for both obese men and obese women in the European labor market, but the weight penalty in wages is more significant for women than for men. Larose et al. [62] found that among working-age adults in Canada, obesity led to larger reductions in hourly wages and annual earnings for women than for men. Caliendo and Gehrsitz [63] analyzed data from Germany showing wage penalties for overweight and obese women in white-collar occupations. In addition, men’s employment prospects increase with weight, and underweight men in blue-collar jobs are paid less. However, Brunello and D’Hombres [64], using data from the European Community Household Panel, observed that the body mass index (BMI) has a greater negative impact on earnings for men than for women. Other researchers, including Behrman and Rosenzweig [65], have not observed any weight penalty in wages for American women. As argued by Cawley [34] and Burkhauser [66], these mixed results may be consequences of different identification strategies and different body size indicators.

Nonetheless, it is important for research on obesity and labor market outcomes to account for gender differences, as policies aimed at preventing obesity will be more effective if they are conveyed in a gender-specific manner. Such policies need to ensure that they do not inadvertently exacerbate excess weight gain or loss, especially as excess-weight penalties vary widely by gender in different economic stages.

For example, in developing countries, Deolalikar [67], using India data, Haddad and Bouis [68], using Philippines data, Thomas and Strauss [69], using Brazil data, Glick and Sahn [70], using Guinea data, and Schultz [71], analyzing data from the Cote d’Ivoire and Ghana, reported a positive and significant effect of body size on both farm outputs and agricultural wage rates. On the other hand, studies conducted in high-income developed countries have shown that weight generally has a negative effect on wages, especially among women, but not among some minority groups [31,32,34,51]. Using data from Finland, Johansson et al. found that waist circumference was negatively associated with women’s wages, but no obesity measure was significant in the linear wage models for men [46]. Assuming a nonlinear wage model, Kropfhauber and Sunder found an inverse u-shaped association between BMI and wages among young German workers [72]. Using data from England, Morris found that BMI had a positive and significant effect on occupational attainment for men and a negative and significant effect for women [37]. Lundborg et al. pooled data from 10 European countries and found a negative correlation between obesity and women’s hourly wages [42]. Table 1 presents a summary of the main research articles related to excess weight and labor market outcomes.

With China’s rapid economic boom and urban expansion, it is not surprising that obesity rates are on the rise, as with urbanization comes more sedentary occupational roles. Along with an increased intake of sugar-laden foods, women, often more physically inactive than men, appear to be more susceptible to the effects of these foods on obesity. Yet, countries with higher incomes and higher female labor force participation rates tend to find that female obesity rates rise more slowly than those of men [4], and that women suffer more pronounced weight penalties in terms of wages than men. The aim of this article is to investigate the excess-weight penalty in wages for men and women in China’s labor market. Despite volumes of research in other parts of the world, only a few have examined the causal relationship from excess weight to employment settings in China, including Shimokawa’s finding of a wage penalty for very heavy and thin persons among both men and women but the penalty was more pronounced for men [73], and Pan, Qin and Liu’s finding of an inverse u-shaped effect of body size on the probability of being employed [75]. Liu et al. found no relationship between income and overweight and obesity in the rural population of Henan province [76].

Here, we put forth three hypotheses:
**Hypothesis 1** **(H1).***There is an inverse u-shaped relationship between excess weight and wages, with employment settings being most favorable when the body size is moderate*.
**Hypothesis 2** **(H2).***Women are more likely than men to be penalized in terms of wages for being overweight or obese*.
**Hypothesis 3** **(H3).**The effect of excess weight on income is related to stages of economic development.

We use a sample of 13,574 urban individuals from nine provinces in the Chinese Household Income Project (CHIP) and regress income on body size. The problem of simultaneity between body mass and labor market outcome and omitted variables complicates our identification. To address this endogeneity issue, we use an instrumental variable approach, i.e., constructing the community-level body mass as our instrumental variable. Our key findings are:There is an inverse u-shape association between BMI and logarithm of monthly income for men, with an implied optimum above the threshold of obesity, while women are better off the slimmer they are;Overweight men received a wage premium and overweight women received a wage penalty compared to their healthy-weight peers;The effect of excess weight on income is most pronounced in Eastern China, which is the most economically developed and commercially active region in China.

We therefore suspect that the excess-weight premium in wages for Chinese urban men is driven by entrenched business practices of excessive dining and drinking associated with senior positions. Food plays a major role in Chinese culture and is often seen as a form of strengthening intimacy between business associates. Not only are formal business meals, often including twelve to sixteen dishes served with white spirits, a key method for establishing and maintaining strong relationships, but it is not unusual for important business decisions to be made at dinners or banquets outside the office [77,78]. Policies aimed at reducing obesity in China must be adapted to its unique sociocultural context in order to have gender-differentiated effects.

The rest of the article is outlined as follows. Section 2 discusses the data and variables, and presents descriptive statistics for the sample. Section 3 presents our regression estimates as well as sensitivity analyses. Section 4 discusses the results, and Section 5 concludes the article.

## 2. Materials and Methods

### 2.1. Data

Our analysis is based on data from the Urban House Survey conducted by China’s National Bureau of Statistics (NBS) as part of the collaborative research project, the Chinese Household Income Project (CHIP), on incomes and inequality in China. The CHIP has carried out several rounds of national household surveys for the reference years 1988, 1995, 1999, 2002, 2007, 2008, and lastly for 2013. Of these, the 2008 urban household survey is a follow-up to the 2007 urban household survey with the same respondents for most of the samples. Specifically, CHIP Urban House Survey 2007 and CHIP Urban House Survey 2008 contain a sample of 5000 urban households, 14,683 individuals for the reference year 2007 and 14,859 individuals for the reference year 2008, from nine provinces: Shanghai, Jiangsu, Zhejiang, and Guangdong from eastern China, Anhui, Henan, and Hubei from central China, and Chongqing and Sichuan from western China.

The nine provinces were selected with the condition that they should constitute a representative sample of the economic characteristics in urban China’s various regions. The respondent households are selected using a two-stage stratified systematic random sampling scheme. In the first stage, all cities and county towns are categorized by population size, divided into geographical regions, and in each region the cities and county towns of each category are arranged according to the average wages of their staff and workers. The number of individuals who are staff and workers in the cities is summed and the sample cities or counties are selected using an interval of one million staff and workers. In the second stage, a multiphase sampling scheme was used to select sample households within the selected cities and county towns, through sample subdistrict and sample resident committees.

The sample was interviewed through questionnaires designed by the project team. Detailed information was collected on household demographics, educational attainments, employment status, health conditions, social and economic characteristics at both personal and household level. We choose to use data from the 2007 and 2008 surveys because they are the only and most up-to-date available surveys that include information on both body size and income; other surveys, including the most recent 2013 CHIP survey, did not collect information on weight and height.

We restrict the sample to the working-age population, i.e., males aged 18 to 60 and females aged 18 to 55, both capped at the official Chinese retirement age. In addition, we only consider samples that had reported information on weight and height. As in previous studies, we restrict the sample to individuals with a BMI below 50 and above 15, thus excluding individual who were very morbidly obese or very severely underweight, but including individuals who were morbid obese or severely underweight. We exclude those samples with extreme BMI values due to concerns about misreported values. We also exclude women who were pregnant or breastfeeding at the time of the survey, or who had given birth in the year right before the survey. Moreover, we exclude self-employed individuals, individuals employed as apprentices, and individuals with disabilities. Individuals in the final sample had to be in full-time employment and had strong labor market ties. Therefore, our final sample consists of 13,574 individuals, of which 7713 are men and 5861 are women, which we treat as unbalanced panel data.

### 2.2. Descriptive Statistics

#### 2.2.1. Dependent Variables

This article examines the effect of excess weight on labor market outcome, which we measure as the logarithm of total monthly income from employment. Employment income is based on individual self-reported values, including salaries and housing and food subsidies from full-time employment.

#### 2.2.2. Independent Variables

Our primary explanatory variable of interest is BMI, defined as weight in kilograms divided by height in meters squared. As anthropometric information in this panel is self-reported, this can lead to BMI measurement errors. Intuitively, individuals tend to under-report their weight and over-report their height. In fact, underreporting of BMI also depends on age and gender, with younger individuals and women more likely to underreport than older individuals and men. Other factors, such as socioeconomic status, may also influence the likelihood of misreporting. Some researchers in the literature, such as Cawley [34] corrected for misreported BMI values using fitted values based on anthropometric heights and weights. In this study, we also have some data on weight and height measurements that were obtained by the interviewers during the survey. We use these measurements to test the true value of BMI and find that the results obtained were very similar to those self-reported values. Therefore, we have no reason to believe that there is significant error in our data due to self-reporting. Another way to address the reporting error is to lower the thresholds for obesity and overweight [73]. Weight in kilograms (controlling for height in meters) and clinical weight classification are also evaluated as our measures of excess weight.

The indicator of underweight is defined as a BMI less than 18.5. The indicator for being overweight is defined as a BMI between 25 and 28, and the indicator for being obese is defined as a BMI of 28 or higher. According to the World Health Organization (WHO), overweight and obesity are defined as 25 ≤ BMI < 30 and BMI ≥ 30, respectively. However, there is growing evidence that the WHO cutpoints derived from the white population may not apply to Asian population, particularly the Chinese working-age population. Since our focus in limited to the impact of body size on labor market performance of the working-age population, here we use overweight and obesity from the Working-Group of Obesity in China (WGOC). The healthy-weight individuals with a BMI between 18.5 and 25 are the control group. In the final sample, around 73% of male and 80% of female participants fall into the normal group, with around 19% of the males and 8% of the females being overweight; the underweight and obese males accounted for 2% and 5% of the sample, respectively, while the underweight and obese females accounted for 9% and 2% of the sample, respectively. Table 2 depicts the descriptive statistics of the participants’ BMI.

We also include an individual’s age, age squared, and whether he or she was married or cohabiting as control variables in the regression. Of these, information on age is obtained by subtracting the individual’s reported year of birth from the year in which the survey was conducted. The analysis also includes individual’s educational level, measured by years of full-time education. Work experience tends to increase earnings due to access to more specialized or higher-skilled jobs. Therefore, we also control for work experience, which is calculated by subtracting the individual’s first year of employment in his or her industry from the survey year. In addition, we control for the year dummy variables in the regression.

We use self-assessed health values to control for health status. Self-assessed health values were measured using a five-point scale in the survey: very good, good, fair, bad, and very bad. Very good health status reports the lowest score (1) and very poor health status reports the highest score (5). We include hukou status as a dummy variable, with 1 indicating urban (non-agricultural) hukou status and 0 indicating rural (agricultural) hukou status. We also consider the ethnicity variable, where 1 indicates majority ethnicity (Han) and 0 indicates minority ethnicity (e.g., Manchu, Hui, etc.). The type of work contract also has an effect on wages, so we treat it as another dummy variable, where 1 denotes permanent or long-term contracts and 0 denotes short-term contracts or other contracts. Table 3 provides the descriptive statistics of our sample.

### 2.3. Method

We use the traditional ordinary least square (OLS) estimiation method to test the effect of body size on larbor market outcome, with other explanatory variables such as age, age squared, education, self-assessed health status, urban hukou status, ethnicity, marital status, workexperience, occupation and industry, and whether the contract is permanent or longterm. The OLS regression is shown below (Equation (1)):(1)$$l_{i}=\alpha +\beta b_{i}+\gamma X_{i}+u_{i},$$
where$\;b_{i}$ is the key explanatory variable body size, and X in equation (1) is the set of control variables that affect an individual’s labor market performance. u is an error term that needs to be uncorrelated over time. Our aim is to estimate β, which determines the impact of an individual’s body size on his or her labor market performance. If b is strictly exogenous, then we can estimate β using the traditional ordinary least square (OLS) estimation.

However, as first explained in detail by Cawley [34], the endogeneity problem of excess weight, which may bias the estimation results, exists for two reasons. First, reverse causality between body size and job market status may lead to endogeneity, as low income or unemployment leads to consumption of cheaper, fatter, higher-calorie foods and thus directly affect body size. Second, other unobserved characteristics of missing variable, such as self-confidence or time preference, may affect both body size and labor market outcome, which may also lead to endogeneity. Thus, we run the risk of overestimating the impact of excess weight on labor market outcome.

Here we adopt a two-stage least squares (2SLS) approach to mitigate endogeneity by using the community level body size prevalence as the instrumental variable (IV). It is expected that the instrumental variable will be correlated with the non-genetic variation of an individual’s excess weight through their impact on food consumption and lifestyle.

## 3. Results

### 3.1. Basic Description

Table 4 summarizes the basic results of the gender-specific income regressions. We use two identification strategies, OLS regression and IV regression, by constructing the community-level body size prevalence as our instrumental variable for each gender.

We find a positive effect of BMI values and a negative effect of BMI squared for men; both are statistically significant at the 5% level. Thus, we observe an inverse u-shaped association between BMI and income for men. This suggests that either being too fat or too slim negatively affect the monthly income of Chinese urban men.

Compared to men, the OLS regression for women reveals a negative and a positive effect of BMI and BMI squared, respectively, both of which are statistically significant at the 5% level. This suggests that as BMI increases, the monthly income of Chinese urban women decreases at an increasing rate. This coincides with the preference for slender females in Chinese culture. However, in the IV regressions for women, both BMI and BMI squared lose their significance.

Before proceeding to the full regressions that include control variables, we must explore the correlations between variables to rule out multicollinearity issue. Table 5 reports the correlation matrix for the key variables. From the table, we can observe that the variables do not have high pair-wise correlations. Therefore, we are able to perform regressions with control variables.

Before proceeding to the multiple regression models with control variables, we use the Variance Inflation Factor (VIF) to test for multicollinearity. Table 6 presents the VIF values for the full regression with different measurement of body size for each gender. The results show that the VIF values are less than 2 for all explanatory variables except BMI and BMI squared (close to 50), weight and weight squared (close to 50), and age and age squared (close to 90). Since we expect a nonlinear relationship between body size and income, it is not reasonable to exclude BMI squared and weight squared. We run the same regression, excluding age squared or replacing age and age squared with log(age), and find that the VIF value for age drops to less than 2, but the results are similar to those that include age squared.

In the benchmark model given in Table 4, the IV regression for women is not significant, which may be due to the endogeneity of body size. The endogeneity problem can be severe if we do not use control variables. However, if we add control variables to the model for regression, the endogeneity problem can be moderated, but still exists. In the following subsections, we use instrumental variables to address the endogeneity problem.

### 3.2. The Effect ofExcess Weight on Income

Table 7 and Table 8 present the results of the wage regression using clinical weight classification as the main explanatory variable. Columns 1, 2, and 3 in each table report the results of the standard OLS estimates, with BMI, weight, and clinical weight classification as the main explanatory variables. Columns 4, 5, and 6 in each table report the results of the 2SLS estimates, using the community-level BMI value, weight prevalence, and weight classification prevalence as instrumental variables. The reference groups include individuals in the healthy BMI range.

As the excess-weight variables are likely to be endogenous, the OLS estimates may be biased. Therefore, we compare the results of OLS and IV estimates. Column 1 in Table 7 shows a positive effect of BMI and a negative effect of BMI squared, and both statistically significant at the 1% level. Therefore, we conclude that the association between BMI and monthly income for men appears to be inverse u-shaped. From the regression results, we can easily calculate the BMI at the peak of the inverse u-shaped curve, i.e., the optimum BMI, which is 30. Column 2 gives the effect of weight (controlling for height) on income, indicating that weight also has an inverse u-shaped effect on income. Column 3 shows the regression results with clinical weight classification as the main explanatory variable. We observe that underweight men are penalized by a 15% wage penalty. In contrast, overweight men receive a 7% wage premium compared to their healthy-weight peers. In addition, obese men do not have lower monthly income compared to healthy-weight men. This finding is consistent with the results in column 1 that obesity is the best choice. Columns 4, 5, and 6 in Table 7 are the results of the IV estimates. The significance of BMI and BMI squared decreases to 5% significance interval, but the effect of BMI value on income is still inverse u-shaped. There is also an inverse u-shaped association between weight and income, with a positive effect of height on income. The effect of underweight men lose significance, yet overweight men have 7.8% extra pay.

Age and its squared value also have a significant effect on monthly income, suggesting that age has a nonlinear effect on income. The coefficients of education and work experience are both significantly positive at the 1% level, indicating that these attributes have positive labor market returns. The effect of hukou status is significantly negative at the 1% level, indicating that men with urban hukou have significantly lower income than men with rural hukou. This may be due to self-selection of the labor force, i.e., those with rural hukou status will stay in the city only if their income is high enough. These individuals will return to rural areas if they are unable to secure high-income jobs. Ethnicity is significantly positive at the 1% level, indicating that Han Chinese have higher incomes than ethnic minorities. Regressions on marital status indicate that married men receive lower income than unmarried men. In addition, having a permanent or a long-term contract has a significant positive effect on income. However, self-assessed health status is not significant in the estimates.

Table 8 shows the results for women. OLS estimates show that the effect of BMI is negative at the 1% significance level, yet, the coefficient of BMI squared lose significance. This demonstrates a negative relationship between BMI and monthly income for Chinese urban women, and thus it can be inferred that the Chinese labor market favors slim women and discriminates against larger women. There is a similar negative effect of weight in the OLS estimates. In addition, height is positively significant at the 1% level. From regressions on clinical weight classifications, we obtain different results for women than for men. We observe that, compared to their healthy-weight peers, underweight women receive 7.2% extra pay, while overweight women received a 4.6% wage penalty. However, the coefficient for obese women is not significant. IV estimation shows similar results for the BMI and weight variables as for men, indicating an inverse u-shaped effect of BMI and weight on women’s monthly income. In contrast to the insignificance in the benchmark model in Table 4, both BMI and BMI squared are significant at the 5% level in the IV regression. This may be due to the endogeneity problem in the benchmark model, which can be addressed to some extent when we add some control variables to the regression. IV estimates based on clinical weight classification shows that both underweight and overweight women receive additional payoffs, but the estimates for obese individuals are not significant.

The effects of age and age squared are different for women than for men. For women, we find a negative effect of age and a positive effect of age squared, suggesting that monthly income declines at an increasing rate as age increases. This may be due to the distribution of our data, as most of our data on women are in the 40 to 55 age range, of which women’s presence in the labor market tends to decrease. The effects of education, work experience and hukou status are similar for women and men. However, the coefficients on ethnicity and marital status are not significant, indicating that the monthly income of Han women is not significantly different from that of ethnic minorities. Moreover, there is no significant difference between married and unmarried females. However, the coefficient for having a permanent or long contract is significantly positive at the 1% level.

### 3.3. The Effect of Excess Weight on Income in Different Regions

As an economy in transition, different regions of China are at different stages of economic development. Some eastern provinces such as Shanghai, Guangdong and Zhejiang have GDP per capita at or near the level of developed economies, presenting a completely different economic landscape from that of the central and western regions. Regressions on the effect of excess weight on income in different regions show that body mass has the greatest effect on income in eastern China. For the least developed western region, the effect of BMI on income is inverse u-shaped but with lower significance, while there is no effect in the central region. Table 9 presents the results of the wage regression for different regions.

### 3.4. The Effect of Excess Weight on Income in Different Sectors

In addition to economic development, occupation may also have differentiated impact on the effect of body size on income. Table 10 presentsan inverse u-shaped relationship between BMI and income in both the secondary and tertiary sectors using OLS regressions, with the optimum BMI being much higher in the secondary sector than in the tertiary sector. This may be due to more manual labor in the secondary sector such as mining, manufacturing and construction. The tertiary sector loses significance using IV regressions.

## 4. Discussion

Unlike most other studies that have found male obesity penalties but not necessarily significant compared to female, our estimates show a significant male obesity premium. Lundborg et al. [42] found that the probability of being employed is significantly lower for both obese men and obese women in the European labor market, but the weight penalty in wages is more significant for women than for men. Shimokawa [73] provide evidence of a wage penalty for very heavy and thin persons among both men and women in China, but the wage penalty was more pronounced for men than for women. The findings in this article are more in line with Morris [37], who pooled cross-sectional health survey data from England and used OLS to find a positive significant effect of BMI on occupational attainment for men and a negative significant effect for women, although the IV coefficients on the BMI measures was not significant in Morris. This article uses OLS and IV regressions to find that BMI has a positive and significant effect on labor market outcomes, both statistically significant at the 5% level. OLS regressions for women reveal a negative effect of BMI, although the IV coefficient on the BMI measure is not significant. We also find an inverse u-shape association between BMI and logarithm of monthly income for men, with an implied optimum, the value of BMI that maximizes earnings ceteris paribus, achieved at a BMI of 30, i.e., above the threshold of obesity, while women are better off the slimmer they are. This finding is consistent with the analysis of China’s data by Pan, Qin and Liu [75].

When we regress income on weights, we use the logarithm of total monthly income from employment as a measure of income, reflecting the changes in income, and look for factors that cause these changes There are two possible channels at work here: reduced productivity due to poor health [50,51,52,53,54,55,56,57], and labor market discrimination that associates weight with personality traits or physical attractiveness [58,59,60,61]. Our regression results suggest that health-induced productivity may not be the main driver when it comes to changes in income, as self-assessed health is not significant in both male and female estimates, a finding that may suggest that discrimination is a larger problem. Similar to most studies investigating the effect of health on overweight penalties (or premiums), we found limited health effects. We suspect that the excess-weight premium in wages for Chinese urban men is driven by entrenched business practices of excessive dining and drinking associated with senior positions. Food plays a major role in Chinese culture and is often seen as a form of strengthening intimacy between business associates. Not only are formal business meals, often including twelve to sixteen dishes served with white spirits, a key method for establishing and maintaining strong relationships, but it is not unusual for important business decisions to be made at dinners or banquets outside the office [77,78]. Therefore, policies aimed at reducing obesity in China must be adapted to its unique socio cultural context in order to have gender-differentiated effects.

We also find that regions (at different economic stages) and occupations have differentiated impact on the effect of body size on earnings. The effect of being overweight on income is most pronounced in the most developed region of China, and more pronounced in the least developed regions as well. In addition, the optimum BMI is significantly higher in the secondary sector than in the other sectors. Brunello and D’Hombres [64] and Lundborg et al. [42] found that the negative effect of excess weight on labor market outcome is more pronounced in relatively underdeveloped Southern and Central Europe than in relatively developed Northern Europe. The current literature points to differences in dietary intake as the main cause of the gender disparities in overweight in developed countries, and differences in reduced physical activity during the shift from agricultural to wage labor as the main cause of the gender disparities in overweight in developing countries. As for China, while the direction and causes of the weight-wage relationship are similar in the most and least developed regions, the underlying culture is very different. In economically developed and commercially active regions, business banquet out of office is key to negotiating business deals, while in least economically developed regions, excessive dining and drinking at official banquets are crucial to establishing power hierarchies or forming alliances. These are particularly pronounced in China’s highly competitive secondary sector, where “guanxi” (relationships) are needed to secure business partnerships.

The limitation of this study arguably lies in the measure of obesity it uses. Contrary to many previous studies, this paper concludes that being overweight is not an issue for men concerning income. However, because our explanatory variable, BMI, tends to underestimate differences in body fat, it underestimates the extent of overweight and obesity. A better indicator would be the percentage of body fat or fat-free mass. In addition, this study is based on data of 2007 and 2008. Although few databases other than CHIP contain both body size and income information in China, and even the most recent CHIP survey in 2013 did not collect information on weight and height, the research data should add up-to-date validation to provide the contribution and reference value. On the other hand, economic development in the central and western regions is catching up to the current level of economic development in the eastern regions. Thus, the effect of BMI on income in the eastern region in 2007 and 2008 is somewhat, though not strictly, representative of the national average in the more recent past. As shown in sub Section 3.3, the effect of BMI on income is most pronounced in eastern China, and thus the rapid economic development of the past decade has not mitigated the wage discrimination in body size. Furthermore, the link between job rank and lifestyle should be better studied, especially for men. In particular, whether China’s anti-corruption campaign helped to reduce weight-wage premium for men, and thus acted to reverse the obesity epidemic deserves further research.

## 5. Conclusions

This article investigates the excess-weight penalty in wages for men and women in China’s labor market. We find a 7% excess-weight premium in wages for overweight men and a 4.6% penalty for overweight women, compared to their healthy-weight peers. We also find an inverse u-shaped association between BMI and logarithm of monthly income for men, with an implied optimum achieved at a BMI of 30, i.e., above the threshold of obesity, while women are better off the slimmer they are. The CHIP survey, initially set up to investigate income and inequality in China, collected information on weight and height for 2007 and 2008, which made possible our study linking body size to wages. Our results are generally consistent with current literature on excess-weight wage penalties in China, including Shimokawa’s finding of a wage penalty for very heavy and thin persons among both men and women but the penalty was more pronounced for men [73], and Pan, Qin and Liu’s finding of an inverse u-shaped effect of body size on the probability of being employed [75]. Our focus, however, is primarily on gender disparities, and the underlying socio cultural complexities. China is on the front lines of the battle against obesity and policies aimed at reducing obesity must be adapted to its unique socio cultural context in order to have gender-differentiated effects. Obesity-related health campaigns alone may exacerbate the already existing discrimination against overweight female, but anti-corruption campaign or, more recently, anti-food-waste campaign may help reduce the weight-wage premium for men, thereby reversing the obesity epidemic and gender disparity. Overall, given the small number of studies to date, the picture of how men are rewarded for being overweight is incomplete. There is clearly a need for further research on occupational attainment and gender disparities in China’s labor market.

## Figures and Tables

**Table 1 ijerph-17-07004-t001:** Main research articles related to excess weight and labor market outcomes.

Reference	Countries/ Regions	Results	
		Relationship between Body Size and Labor Market Outcomes	Gender Disparities
Averett and Korenman [32]	United States	There are economic penalties to being overweight, but the penalty is much smaller among black women.	Obese women have lower family incomes than women whose weight-for-height is in the “recommended” range, while results for men are weaker and mixed.
Cawley [34,51]	United States	Both body mass and weight have negative wage effect, but the significance of this effect is not obvious among Hispanic workers.	The negative impact of body mass and weight on wages is largest for white females and smallest for black females.
Larose et al. [62]	Canada	Among working-age adults, obesity led to larger reductions in hourly wages and annual earnings for women than for men.	
Brunello and D’Hombres [64]	Pooling 9 European countries	Body mass has a negative effect on earnings, and the impact is larger and statistically more significant in Southern Europe than in Northern Europe.	A 10% increase in the average BMI reduces the real earnings of men and women by 3.27% and 1.86%, respectively.
Lundborg et al. [42]	Pooling 10 European countries	Obesity has significant negative effect on the probability of being employed, and the effect is most pronounced for men in Southern and Central Europe.	Obese European women earned 10% less than their non-obese peers, while for men the effect is smaller and insignificant. Obese women in central Europe faced the greatest wage penalty.
Sargent and Blanchflower [31]	Great Britain	There is a statistically significant inverse relationship between obesity and earnings for women, while there are no obesity effects for men.	
Morris [37]	England	BMI has a negative and significant effect on occupational attainment in women, while the results for men are mixed.	
Kropfhauberand Sunder [72]	Germany	There is an inverse u-shaped relationship between BMI and log wages.	The optimum BMI for wage is achieved at 30 for men and 27 for women.
Greve [45]	Denmark	In the private sector, BMI has a negative effect on wages for women but an inverse u-shaped effect on wages for men, whereas results from the public sector show that BMI has no influence on wages for either men or women.	
Johansson et al. [46]	Finland	Waist circumference has a negative association with wages for women, whereas no obesity measure is significant in the linear wage models for men. Obesity is negatively associated with women’s employment probability and fat mass is negatively associated with men’s employment probability.	
Dackehag et al. [73]	Sweden	There is a significant obesity penalty in income for men, but no significant excess-weight penalty for women.	
Haddad and Bouis [68]	A southern Philippine province	There is a positive relationship between body size and labor productivity as measured by agricultural wages.	N/A
Thomas and Strauss [69]	Brazil	Health measures, such as BMI and per capita calorie intake, positively and significantly affected wages.	BMI affected only men’s wage.
Shimokawa [74]	China	There is a wage penalty for very heavy and thin persons.	The wage penalty is more significant among men than among women.
Pan et al. [75]	China	Body size has an inverse u-shaped effect on the probability of being employed.	The optimal BMI for employment is estimated to be 24.3 for male and 22.7 for female.

**Table 2 ijerph-17-07004-t002:** Proportions of clinical classifications in the CHIP sample.

	2007		2008	
	Male	Female	Male	Female
Underweight (BMI < 18.5)	79 (2.1%)	274(9.4%)	88 (2.3%)	270 (9.2%)
Normal (18.5 ≤ BMI < 25)	2903 (74.8%)	2356 (80.4%)	2758 (72.0%)	2334 (79.6%)
Overweight (25 ≤ BMI < 28)	708 (18.2%)	241 (8.2%)	773 (20.2%)	271 (9.3%)
Obese (BMI ≥ 28)	192 (4.9%)	59 (2.0%)	212 (5.5%)	56 (1.9%)
Total	3882	2930	3831	2931

Note: BMI, body mass index.

**Table 3 ijerph-17-07004-t003:** Descriptive statistics.

	Male				Female			
	Mean	Standard Deviation	Min	Max	Mean	Standard Deviation	Min	Max
Observations	6972				5861			
Income_hour	17.55	25.99	0.00	1000.00	13.11	17.64	0.00	562.50
BMI	23.42	2.83	14.69	48.44	21.71	2.74	15.06	46.29
Age	41.85	10.38	18.00	60.00	38.54	9.13	18.00	55.00
Education	12.85	3.01	0.00	19.00	12.73	2.92	0.00	19.00
Experience	14.92	11.30	0.00	43.00	11.33	9.52	0.00	38.00
Health	2.12	0.69	1.00	5.00	2.15	0.68	1.00	5.00
Urban hukou status	0.97	0.17	0.00	1.00	0.96	0.20	0.00	1.00
Ethnicity	0.99	0.10	0.00	1.00	0.99	0.10	0.00	1.00
Marriage status	0.14	0.34	0.00	1.00	0.13	0.33	0.00	1.00
Long-term contract	0.85	0.36	0.00	1.00	0.78	0.41	0.00	1.00

**Table 4 ijerph-17-07004-t004:** Income and body size.

	Male		Female	
Variable Name	Ordinary Least Square Estimation (OLS)	Instrumental Variable Estimation (IV)	OLS	IV
Body mass index (BMI)	0.107 ***	0.147 **	−0.069 **	0.063
	(4.41)	(2.17)	(−4.14)	(1.3)
Square of BMI	−0.002 ***	−0.003 ***	0.001 ***	−0.001
	(−3.70)	(−2.18)	(2.80)	(−1.45)
R^2^	0.007	0.002	0.012	

Note: t statistics in parentheses ** *p* < 0.05, *** *p* < 0.01.

**Table 5 ijerph-17-07004-t005:** Correlation matrix of variables

	BMI	Age	Education	Experience	Health	Year Dummy	Urban Hukou Status	Ethnicity	Marital Status	Long-Term Contract
BMI	1.00									
Age	0.25	1.00								
Education	−0.06	−0.24	1.00							
Experience	0.15	0.48	0.00	1.00						
Health	0.03	0.21	−0.07	0.05	1.00					
Year Dummy	0.01	0.02	0.05	0.05	0.08	1.00				
Urban hukou status	0.02	0.06	0.09	0.06	0.03	0.02	1.00			
Ethnicity	0.00	0.00	−0.01	0.00	−0.02	0.00	−0.01	1.00		
Marital status	−0.17	−0.54	0.12	−0.39	−0.11	0.00	0.03	0.01	1.00	
long-term contract	0.02	−0.04	0.22	0.31	−0.05	0.03	0.08	0.03	−0.03	1.00

**Table 6 ijerph-17-07004-t006:** The Variance Inflation Factor (VIF).

Variables	Male			Female		
Body mass index (BMI)	52.14			52.14		
Square of BMI	51.74			51.74		
Weight			49.9		45.93	
Square of weight			48.5		44.31	
Height			1.37		1.31	
Dummy, =1 if overweight		1.07				1.08
Dummy, =1 if obese		1.04				1.02
Dummy, =1 if underweight		1.03				1.07
Age	91.98	92.08	92.03	91.98	79.37	79.05
Square of Age	81.71	81.83	81.69	81.71	72.05	71.99
Dummy, =1 if married	2.07	2.07	2.08	2.07	1.78	1.77
Log (occuyear)	1.65	1.65	1.65	1.65	1.51	1.51
Dummy, =1 if permanent or long-term contract	1.18	1.18	1.18	1.18	1.22	1.22
Log (edu)	1.11	1.11	1.11	1.11	1.19	1.18
Health status	1.07	1.07	1.07	1.07	1.05	1.05
Dummy, =1 if urban hukou	1.02	1.02	1.02	1.02	1.04	1.04
Year dummy	1.01	1.01	1.01	1.01	1.01	1.01
Dummy, =1 if Han ethnic	1	1	1	1	1	1

**Table 7 ijerph-17-07004-t007:** Wage regression for men.

	Ordinary Least Square Estimation (OLS)			Instrumental Variable Estimation (IV)		
	(1)	(2)	(3)	(4)	(5)	(6)
Body mass index (BMI)	0.060 ***			0.121 **		
	(2.79)			(2.12)		
Square of BMI	−0.001 ***			−0.002 **		
	(−2.20)			(−2.14)		
Weight		0.017 ***			0.063 **	
		(2.74)			(1.93)	
Square of Weight		−0.000 **			−0.000 **	
		(−2.07)			(−2.10)	
Height		0.001			0.007**	
		(0.30)			(2.16)	
Dummy, =1 if underweight			−0.151 **			0.400
			(2.50)			(1.59)
Dummy, =1 if overweight			0.071 ***			0.078 *
			(4.44)			(1.72)
Dummy, =1 if obese			0.056			−0.000
			(1.63)			(−0.01)
Age	0.014 *	0.015 *	0.014 *	0.014 *	0.017 **	0.020 **
	(1.75)	(1.88)	(1.80)	(1.74)	(1.97)	(2.36)
Square of Age	−0.000 ***	−0.000 ***	−0.000 ***	−0.000 ***	−0.000 ***	−0.000 ***
	(−2.91)	(−2.99)	(−2.96)	(−2.87)	(−3.03)	(−3.40)
Log (Edu)	0.598 ***	0.593 ***	0.597 ***	0.595 ***	0.587 ***	0.602 ***
	(13.63)	(13.63)	(13.57)	(13.59)	(13.56)	(13.61)
Log (Experience)	0.144 ***	0.143 ***	0.145 ***	0.144 ***	0.142 ***	0.144 ***
	(13.99)	(13.91)	(14.02)	(13.92)	(13.51)	(13.84)
Health status	0.008	0.008	0.007	0.009	0.011	0.003
	(0.70)	(0.73)	(0.65)	(0.79)	(0.95)	(0.29)
Year dummy	0.101 ***	0.102 ***	0.102***	0.100 ***	0.100 ***	0.101 ***
	(6.85)	(6.89)	(6.95)	(6.78)	(6.63)	(6.82)
Dummy, =1 if urban hukou	−0.165 ***	−0.171 ***	−0.163 ***	−0.165 ***	−0.173 ***	−0.179 ***
	(−3.64)	(−3.76)	(−3.58)	(−3.61)	(−3.74)	(−3.91)
Dummy, =1 if Han ethnic	0.240 ***	0.236 ***	0.236 ***	0.240 ***	0.236 ***	0.236 ***
	(3.10)	(3.05)	(3.06)	(3.13)	(3.11)	(3.09)
Dummy, =1 if married	−0.073 **	−0.081 **	−0.077 **	−0.077 **	−0.088 **	−0.074 **
	(−2.32)	(−2.54)	(−2.46)	(−2.41)	(−2.62)	(−2.32)
Dummy, =1 if permanent or long-term contract	0.374 ***	0.371 ***	0.376 ***	0.376 ***	0.374 ***	0.379 ***
	(16.14)	(15.98)	(16.22)	(16.16)	(15.87)	(16.13)
constant	4.390 ***	4.406 ***	5.239 ***	3.771 ***	1.987 *	5.105 ***
	(12.33)	(11.69)	(23.84)	(5.29)	(1.93)	(21.84)
N	6894	6894	6894	6894	6894	6894
R^2^	0.213	0.213	0.213	0.208	0.187	0.198

Note: t statistics in parentheses * *p* < 0.1, ** *p* < 0.05, *** *p* < 0.01.

**Table 8 ijerph-17-07004-t008:** Wage regression for women.

	Ordinary Least Square Estimation (OLS)			Instrumental Variable Estimation (IV)		
	(1)	(2)	(3)	(4)	(5)	(6)
Body mass index (BMI)	−0.034 ***			0.100 **		
	(−2.13)			(2.06)		
Square of BMI	0.000			−0.002 **		
	(1.49)			(−2.08)		
Weight		−0.010 *			0.034 **	
		(−1.87)			(2.13)	
Square of Weight		0.000			−0.001 **	
		(1.13)			(−2.07)	
Height		0.007 ***			−0.000	
		(3.83)			(−0.04)	
Dummy, =1 if underweight			0.072 ***			0.225 ***
			(2.63)			(3.15)
Dummy, =1 if overweight			−0.046 **			0.180 **
			(−2.03)			(2.17)
Dummy, =1 if obese			−0.077			0.359
			(−1.35)			(1.31)
Age	−0.035 ****	−0.034 ***	−0.036 ***	−0.040 ***	−0.043 ***	−0.034 ***
	(−3.74)	(−3.69)	(−3.86)	(−4.27)	(−4.30)	(−3.69)
Square of Age	0.000 ***	0.000 ***	0.000 ***	0.000 ***	0.000 ***	0.000 ***
	(2.74)	(2.69)	(2.81)	(3.08)	(3.17)	(2.45)
Log (Edu)	0.504 ***	0.500 ***	0.507 ***	0.516 ***	0.508 ***	0.520 ***
	(8.84)	(8.73)	(8.86)	(8.94)	(8.72)	(8.98)
Log (Experience)	0.168 ***	0.167 ***	0.168 ***	0.169 ***	0.168 ***	0.174 ***
	(15.65)	(15.58)	(15.68)	(15.69)	(15.02)	(15.74)
Health status	0.007	0.008	0.008	0.008	0.010	0.001
	(0.59)	(0.69)	(0.64)	(0.64)	(0.78)	(0.04)
Year dummy	0.101 ***	0.101 ***	0.101 ***	0.098 ***	0.097 ***	0.098 ***
	(6.36)	(6.35)	(6.35)	(6.14)	(5.88)	(6.10)
Dummy, =1 if urban hukou	−0.087 **	−0.088 **	−0.085 **	−0.091 **	−0.089 **	−0.089 **
	(−2.41)	(−2.44)	(−2.35)	(−2.50)	(−2.38)	(−2.43)
Dummy, =1 if Han ethnic	0.103	0.109	0.103	0.098	0.089	0.096
	(1.20)	(1.27)	(1.20)	(1.12)	(1.01)	(1.07)
Dummy, =1 if married	−0.041	−0.046	−0.041	−0.030	−0.033	−0.044
	(−1.25)	(−1.41)	(−1.26)	(−0.89)	(−0.94)	(−1.31)
Dummy, =1 if permanent or long-term contract	0.325 ***	0.322 ***	0.325 ***	0.322 ***	0.317 ***	0.326 ***
	(14.80)	(14.70)	(14.80)	(14.65)	(14.03)	(14.66)
constant	6.764 ***	5.506 ***	6.266 ***	5.156 ***	2.507	6.184 ***
	(21.42)	(14.35)	(24.24)	(8.42)	(1.48)	(23.69)
N	5329	5329	5329	5329	5329	5329
R^2^	0.219	0.219	0.218	0.209	0.158	0.195

Note: t statistics in parentheses * *p* < 0.1, ** *p* < 0.05, *** *p* < 0.01.

**Table 9 ijerph-17-07004-t009:** Wage regression for different regions.

	(1)	(2)	(3)	(4)	(5)	(6)
	OLS Eastern Region	OLS Central Region	OLS Western Region	IV Eastern Region	IV Central Region	IV Western Region
Body mass index (BMI)	0.075 ^***^	0.042 ^*^	0.037	0.192 ^***^	0.086	0.531 ^*^
	(3.97)	(1.77)	(0.87)	(3.68)	(0.84)	(1.89)
Square of BMI	−0.001 ^***^	−0.001	−0.000	−0.004 ^***^	−0.002	−0.011 ^*^
	(−3.19)	(−1.33)	(−0.47)	(−3.65)	(−1.01)	(−1.86)
Age	−0.009	−0.017	0.013	−0.010	−0.016	0.005
	(−1.18)	(−1.31)	(1.22)	(−1.30)	(−1.12)	(0.39)
Square of Age	−0.000	0.000	−0.000	−0.000	0.000	−0.000
	(−0.24)	(0.74)	(−1.59)	(−0.15)	(0.71)	(−0.82)
Log (edu)	0.588 ^***^	0.493 ^***^	0.706 ^***^	0.582 ^***^	0.491 ^***^	0.715 ^***^
	(11.36)	(9.05)	(8.44)	(11.33)	(9.02)	(8.60)
Log (occuyear)	0.190 ^***^	0.177 ^***^	0.147 ^***^	0.190 ^***^	0.175 ^***^	0.149 ^***^
	(18.95)	(12.41)	(9.94)	(18.92)	(12.03)	(9.75)
Health status	−0.037 ^***^	−0.022	−0.044 ^***^	−0.035 ^***^	−0.021	−0.040 ^**^
	(−3.31)	(−1.48)	(−2.76)	(−3.10)	(−1.41)	(−2.44)
Year dummy	0.094 ^***^	0.083 ^***^	0.158 ^***^	0.091 ^***^	0.081 ^***^	0.161 ^***^
	(6.26)	(4.55)	(7.33)	(6.04)	(4.41)	(7.22)
Dummy, =1 if urban hukou	−0.045	0.009	0.020	−0.039	0.012	0.022
	(−1.33)	(0.14)	(0.29)	(−1.16)	(0.18)	(0.30)
Dummy, =1 if Han ethnic	−0.109	0.212 ^***^	0.221 ^***^	−0.112	0.193 ^***^	0.204 ^**^
	(−1.14)	(3.31)	(2.78)	(−1.19)	(2.98)	(2.31)
Dummy, =1 if married	−0.058 ^**^	−0.087 ^**^	0.089 ^*^	−0.058 ^**^	−0.092 ^**^	0.104 ^**^
	(−2.01)	(−2.04)	(1.89)	(−1.98)	(−2.13)	(2.12)
Dummy, =1 if permanent or long-term contract	0.316 ^***^	0.415 ^***^	0.291 ^***^	0.314 ^***^	0.429 ^***^	0.284 ^***^
	(14.98)	(15.20)	(8.28)	(14.83)	(15.59)	(7.65)
constant	5.118 ^***^	4.952 ^***^	3.956 ^***^	3.836 ^***^	4.652 ^***^	−1.412
	(15.71)	(10.74)	(7.13)	(5.96)	(3.86)	(−0.46)
*N*	6121	3639	2463	6121	3639	2463
adj. *R*^2^	0.232	0.262	0.268	0.226	0.246	0.220

Note: t statistics in parentheses * *p* < 0.1, ** *p* < 0.05, *** *p* < 0.01.

**Table 10 ijerph-17-07004-t010:** Wage regression for different industries.

	(1)	(2)	(3)	(4)	(5)	(6)
	OLS Primary Industry	OLS Secondary Industry	OLS Tertiary	IV Primary Industry	IV Secondary Industry	IV Tertiary
Body mass index (BMI)	−0.008	0.076 ^***^	0.042 ^**^	0.163	0.101 ^**^	0.185
	(−0.04)	(3.05)	(2.52)	(0.09)	(2.11)	(1.45)
Square of BMI	0.001	−0.001 ^**^	−0.001 ^*^	−0.001	−0.002 ^**^	−0.004
	(0.25)	(−2.31)	(−1.83)	(−0.04)	(−2.47)	(−1.40)
Age	−0.068	−0.020 ^**^	−0.009	−0.124	−0.018 ^*^	−0.011
	(−0.94)	(−2.04)	(−1.16)	(−1.40)	(−1.78)	(−1.36)
Square of Age	0.001	0.000 ^*^	−0.000	0.001	0.000	0.000
	(1.14)	(1.69)	(−0.11)	(1.43)	(1.53)	(0.17)
Log (edu)	0.594 ^*^	0.542 ^***^	0.579 ^***^	0.583 ^**^	0.542 ^***^	0.576 ^***^
	(1.95)	(9.00)	(13.16)	(2.10)	(8.95)	(13.12)
Log (occuyear)	0.006	0.074 ^***^	0.203 ^***^	0.019	0.072 ^***^	0.202 ^***^
	(0.08)	(4.81)	(22.82)	(0.24)	(4.72)	(22.65)
Health status	−0.033	0.018	−0.009	−0.063	0.018	−0.007
	(−0.38)	(1.08)	(−0.88)	(−0.62)	(1.12)	(−0.67)
Year dummy	0.289 ^**^	0.115 ^***^	0.090 ^***^	0.241 ^**^	0.113 ^***^	0.090 ^***^
	(2.49)	(5.16)	(7.12)	(2.00)	(5.05)	(6.99)
Dummy, =1 if urban hukou	0.176	−0.273 ^***^	−0.076 ^**^	0.102	−0.271 ^***^	−0.074 ^**^
	(0.56)	(−4.51)	(−2.26)	(0.26)	(−4.44)	(−2.18)
Dummy, =1 if Han ethnic	0.964 ^***^	−0.114	0.228 ^***^	0.885 ^**^	−0.118	0.228 ^***^
	(3.11)	(−0.85)	(3.57)	(2.12)	(−0.89)	(3.58)
Dummy, =1 if married	−0.281	−0.077 ^*^	0.004	−0.465	−0.077 ^*^	0.005
	(−0.87)	(−1.66)	(0.16)	(−1.38)	(−1.65)	(0.18)
Dummy, =1 if permanent or long-term contract	0.297	0.321 ^***^	0.382^***^	0.305	0.325 ^***^	0.381 ^***^
	(1.34)	(9.06)	(20.89)	(1.41)	(9.16)	(20.56)
constant	5.607 ^**^	5.395 ^***^	4.847 ^***^	4.546	5.208 ^***^	3.293 ^**^
	(2.44)	(13.00)	(16.63)	(0.22)	(7.79)	(2.30)
*N*	118	3285	8820	118	3285	8820
adj. *R*^2^	0.188	0.129	0.257	0.123	0.123	0.249

Note: t statistics in parentheses * *p* < 0.1, ** *p* < 0.05, *** *p* < 0.01.

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
