# Peer review of "Wage Penalties or Wage Premiums? A Socioeconomic Analysis of Gender Disparity in Obesity in Urban China"

_ijerph, 2020, doi:10.3390/ijerph17197004_

Round 1

Reviewer 1 Report

It is a very interesting topic, especially in China, where advanced development and economic prosperity are important topics of concern. This article submitted on " International Journal of Environmental Research and Public Health (ISSN 1660-4601; ISSN 1661-7827) ". This is a world-class, top journal and is included in the Journal of High Visibility in the Science Citation Index and Social Sciences Citation Index. Therefore, the internal text published is advanced and innovative, as well as a certain degree of contribution. It is therefore suggested that the following should be considered for publication after specific changes have been made. The specific recommendations are as follows:

  1. In the past decade, China's economy of the high grow has attracted global attention, this study is based on the data of 2007 and 2008. The GDP has changed greatly in recent years, so it is recommended that the research data should increase the latest validation data of 2017 and 2018 at least, compared with the verification data of 2007 and 2008 to provide the contribution and reference value of the article.

  1. From a scientific point of view, the sampling method must be clearly described in the paper. In the course of the review, it is not appropriate to obviously record the sponsors and partners of the research in the articles submitted. It is recommended that the text to be included in the 2. Materials and Methods 2.1. Data content, “The survey was conducted by the Australian National University and Beijing Normal University and sponsored by the China National Bureau of Statistics (NBS) and the Institute for the Study of Labor (IZA).”. Please delete this passage, please.

  1. Before establishing the multiple regression models (table 4 and table 5), the authors should discuss the correlation between variables and suggest listing the results of the "Correlations Matrix". For example, the correlation between Age and BMI.

  1. The possible problems (proof) of multiple regression models established in table 4 and table 5 should be excluded before it is recommended to use the variance inflation factor to a test of the multicollinearity. Multiple regression models established in table 4 and table 5 should first be excluded from problem (proof), so it is recommended to use the variance inflation factor to a test of the multicollinearity. For example, In Table 3, the Female’s IV model estimated parameter is not a significant. However, in the case of the BMI estimation parameters of the model (4) in Table 5 becoming significant, this may be the result of spurious regression.

  1. Add relevant literature discussions beyond 2016. There are only three relevant articles in this paper after 2016:
  • Tartof, S.; Qian, L.; Wei, R. et al. Obesity and mortality among patients diagnosed with COVID-19: results 382 from an integrated health care organization. Ann. Intern. Med. 2020. https://doi.org/10.7326/M20-3742
  • Zhang, X.; Zhang, M.; Zhao, Z. et al. Geographic variation in prevalence of adult obesity in China: results from the 2013–2014 National Chronic Disease and Risk Factor Surveillance. Ann. Intern. Med. 2020, 172(4),347 291-293.
  • Kim, T.J.; von dem Knesebeck, O. Income and obesity: what is the direction of the relationship? A systematic review and meta-analysis. BMJ Open 2018, 8, e019862. doi:10.1136/bmjopen-2017-019862.

Only (2) and (3) of the above documents are relevant to this study, and it is recommended to add the latest literatures.

  1. It is recommended to take into account the "Province" (inspired by the Zhang et al., 2020 literature), the Genetics are incorporated into the BMI's interpreting variables, and the effect of occupational differences on total monthly income from employment.

Author Response

Response to Reviewer 1 Comments

Point 1: It is a very interesting topic, especially in China, where advanced development and economic prosperity are important topics of concern. This article submitted on " International Journal of Environmental Research and Public Health (ISSN 1660-4601; ISSN 1661-7827) ". This is a world-class, top journal and is included in the Journal of High Visibility in the Science Citation Index and Social Sciences Citation Index. Therefore, the internal text published is advanced and innovative, as well as a certain degree of contribution. It is therefore suggested that the following should be considered for publication after specific changes have been made.

Thank you for your careful evaluation of the manuscript. We sent the revised manuscript according to suggestions and answers to the questions below.

The specific recommendations are as follows:

In the past decade, China's economy of the high grow has attracted global attention, this study is based on the data of 2007 and 2008. The GDP has changed greatly in recent years, so it is recommended that the research data should increase the latest validation data of 2017 and 2018 at least, compared with the verification data of 2007 and 2008 to provide the contribution and reference value of the article.

Response 1: Thanks for the comment. For this manuscript, unfortunately, we were only able to use data of 2007 and 2008. The Chinese Household Income Project (CHIP) surveys, initially set up to investigate income and inequality in China, collected information on weight and height for 2007 and 2008, which made possible our study linking body size to wages. Other rounds of surveys, including the most recent 2013 survey, did not collect information on weight and height. We recognize this point in the limitations of the study, as presented in the manuscript in the “discussion” section:

[line 413-416]

“In addition, this study is based on data of 2007 and 2008. Although few databases other than CHIP contain both body size and income information, and even the most recent CHIP survey in 2013 did not collect information on weight and height, the research data should add up-to-date validation to provide the contribution and reference value.”

Point 2: From a scientific point of view, the sampling method must be clearly described in the paper. In the course of the review, it is not appropriate to obviously record the sponsors and partners of the research in the articles submitted. It is recommended that the text to be included in the 2. Materials and Methods 2.1. Data content, “The survey was conducted by the Australian National University and Beijing Normal University and sponsored by the China National Bureau of Statistics (NBS) and the Institute for the Study of Labor (IZA).”. Please delete this passage, please.

Response 2: Thanks for the comment. We have added the sampling methods to our manuscript in subsection “Data” of section “Materials and Methods”.

[line 152-161]

“The nine provinces were selected with the condition that they should constitute a representative sample of the economic characteristics in urban China’s various regions. The respondent households are selected using a two-stage stratified systematic random sampling scheme. In the first stage, all cities and county towns are categorized by population size, divided into geographical regions, and in each region the cities and county towns of each category are arranged according to the average wages of their staff and workers. The number of individuals who are staff and workers in the cities is summed and the sample cities or counties are selected using an interval of one million staff and workers. In the second stage, a multiphase sampling scheme was used to select sample households within the selected cities and county towns, through sample subdistrict and sample resident committees.

The sample was interviewed through questionnaires designed by the project team. Detailed information was collected on household demographics, educational attainments, employment status, health conditions, social and economic characteristics at both personal and household level. We choose to use data from the 2007 and 2008 surveys because these are the only surveys that include information on body size; other surveys, including the most recent 2013 survey, did not collect information on weight and height.”

We apologize for improperly recording the CHIP sponsorship and partnership, which we have deleted from the manuscript.

Point 3: Before establishing the multiple regression models (table 4 and table 5), the authors should discuss the correlation between variables and suggest listing the results of the "Correlations Matrix". For example, the correlation between Age and BMI.

Response 3: Thanks for the suggestion. We have included the correlation matrix to our manuscript in subsection “Basic Description” of section “Results”.

[line 265-270]

“Before proceeding to the full regressions that include control variables, we must explore the correlations between variables to rule out multicollinearity issue. Table 5 reports the correlation matrix for the key variables. From the table, we can observe that the variables do not have high pair-wise correlations. Therefore, we are able to perform regressions with control variables.”

Table 5. Correlation matrix of variables

BMI

age

education

experience

health

D_year

Urban hukou status

ethnicity

marital status

long-term contract

BMI

1.00

age

0.25

1.00

education

-0.06

-0.24

1.00

experience

0.15

0.48

0.00

1.00

health

0.03

0.21

-0.07

0.05

1.00

D_year

0.01

0.02

0.05

0.05

0.08

1.00

Urban hukou status

0.02

0.06

0.09

0.06

0.03

0.02

1.00

ethnicity

0.00

0.00

-0.01

0.00

-0.02

0.00

-0.01

1.00

Marital status

-0.17

-0.54

0.12

-0.39

-0.11

0.00

0.03

0.01

1.00

long-term contract

0.02

-0.04

0.22

0.31

-0.05

0.03

0.08

0.03

-0.03

1.00

Point 4: The possible problems (proof) of multiple regression models established in table 4 and table 5 should be excluded before it is recommended to use the variance inflation factor to a test of the multicollinearity. Multiple regression models established in table 4 and table 5 should first be excluded from problem (proof), so it is recommended to use the variance inflation factor to a test of the multicollinearity. For example, In Table 3, the Female’s IV model estimated parameter is not a significant. However, in the case of the BMI estimation parameters of the model (4) in Table 5 becoming significant, this may be the result of spurious regression.

Response 4: Thanks for the suggestion. We have checked the multicollinearity using VIF and included to our manuscript in subsection “Basic Description” of section “Results”.

[line 271-276]

“We use Variance Inflation Factor (VIF) to test for multicollinearity. The results show that all explanatory variables have VIF values less than 2, except for BMI and BMI squared (close to 50), and age and age squared (close to 90). Since we expect a nonlinear relationship between BMI and income, it is not reasonable to exclude BMI squared. We run the same regression, excluding age squared, and find that the VIF value for age decreases to less than 2. However, the results are similar to those that include age squared.”

Point 5:

    Add relevant literature discussions beyond 2016. There are only three relevant articles in this paper after 2016:

    Tartof, S.; Qian, L.; Wei, R. et al. Obesity and mortality among patients diagnosed with COVID-19: results 382 from an integrated health care organization. Ann. Intern. Med. 2020. https://doi.org/10.7326/M20-3742

    Zhang, X.; Zhang, M.; Zhao, Z. et al. Geographic variation in prevalence of adult obesity in China: results from the 2013–2014 National Chronic Disease and Risk Factor Surveillance. Ann. Intern. Med. 2020, 172(4),347 291-293.

Kim, T.J.; von dem Knesebeck, O. Income and obesity: what is the direction of the relationship? A systematic review and meta-analysis. BMJ Open 2018, 8, e019862. doi:10.1136/bmjopen-2017-019862.

Only (2) and (3) of the above documents are relevant to this study, and it is recommended to add the latest literatures.

Response 5: Thanks for the recommendation. We have added eight recent articles to our literature discussions, three of which we provide specific discussion (line 67-71, 109-110).

[5] Ogden, C.L.; Fakhouri, T.H.; Carroll, M. D. et al. Prevalence of obesity among Adults, by household income and education — United States, 2011–2014. Morb. Mortal. Wkly. Rep. 2017, 66(50), 1369–1373.

[28] Goisis, A.; Sacker, A.; Kelly, Y. Why are poorer children at higher risk of obesity and overweight? A UK cohort study. Eur. J. Public Health 2016, 26(1), 7-13.

[47] Böckerman, P.; Cawley, J.; Viinikainen, J. et al. The effect of weight on labor market outcomes: An application of genetic instrumental variables. Health Econ. 2019, 28(1), 65-77.

[48] Tyrrell, J.; Jones, S.E.; Beaumont, R. et al. Height, body mass index, and socioeconomic status: Mendelian randomisation study in UK Biobank. BMJ 2016, 352, i582.

[49] Bao, Y.; Clarke, P.S.; Smart, M.; Kumari, M. Assessing the robustness of sisVIVE in a Mendelian randomization study to estimate the causal effect of body mass index on income using multiple SNPs from understanding society. Stat. Med. 2018, 38(9), 1529-1542.

[62] Larose, S.L.; Kpelitse, K.A.; Campbell, M.K. et al. Does obesity influence labour market outcomes among working-age adults? Evidence from Canadian longitudinal data. Econ. Hum. Biol. 2016, 20, 26–41.

[63] Caliendo, M.; Gehrsitz, M. Obesity and the labor market: A fresh look at the weight penalty. Econ. Hum. Biol. 2016, 23, 209–225.

[75] Liu, X.; Wu, W.; Mao, Z. et al. Prevalence and influencing factors of overweight and obesity in a Chinese rural population: the Henan Rural Cohort Study. Sci. Rep. 2018, 8, 13101.

Point 6: It is recommended to take into account the "Province" (inspired by the Zhang et al., 2020 literature), the Genetics are incorporated into the BMI's interpreting variables, and the effect of occupational differences on total monthly income from employment.

Response 6: Thanks for the recommendation. We have examined the effect of excess weight on income in different regions as well as different sectors and included to our manuscript as section 3.3 and section 3.4.

[line 338-357]

3.3. The Effect of Excess Weight on Income in different regions

As an economy in transition, different regions of China are at different stages of economic development. Some eastern provinces such as Shanghai, Guangdong and Zhejiang have GDP per capita at or near the level of developed economies, presenting a completely different economic landscape from that of the central and western regions. Regressions on the effect of excess weight on income in different regions show that excess weight has the greatest effect on income in eastern China. For the least developed western region, the effect of BMI on income is inverse u-shaped but with lower significance, while there is no effect in the central region. Table 8 presents the results of the wage regression for different regions.

Table 8. Wage regression for different regions.

(1)

(2)

(3)

(4)

(5)

(6)

ols_east

ols_middle

ols_west

IV_east

IV_middle

IV_west

bmi

0.075***

0.042*

0.037

0.192***

0.086

0.531*

(3.97)

(1.77)

(0.87)

(3.68)

(0.84)

(1.89)

bmi2

-0.001***

-0.001

-0.000

-0.004***

-0.002

-0.011*

(-3.19)

(-1.33)

(-0.47)

(-3.65)

(-1.01)

(-1.86)

age

-0.009

-0.017

0.013

-0.010

-0.016

0.005

(-1.18)

(-1.31)

(1.22)

(-1.30)

(-1.12)

(0.39)

age2

-0.000

0.000

-0.000

-0.000

0.000

-0.000

(-0.24)

(0.74)

(-1.59)

(-0.15)

(0.71)

(-0.82)

ln_edu

0.588***

0.493***

0.706***

0.582***

0.491***

0.715***

(11.36)

(9.05)

(8.44)

(11.33)

(9.02)

(8.60)

ln_occuyear

0.190***

0.177***

0.147***

0.190***

0.175***

0.149***

(18.95)

(12.41)

(9.94)

(18.92)

(12.03)

(9.75)

health

-0.037***

-0.022

-0.044***

-0.035***

-0.021

-0.040**

(-3.31)

(-1.48)

(-2.76)

(-3.10)

(-1.41)

(-2.44)

D_year

0.094***

0.083***

0.158***

0.091***

0.081***

0.161***

(6.26)

(4.55)

(7.33)

(6.04)

(4.41)

(7.22)

D_hukou

-0.045

0.009

0.020

-0.039

0.012

0.022

(-1.33)

(0.14)

(0.29)

(-1.16)

(0.18)

(0.30)

D_nation

-0.109

0.212***

0.221***

-0.112

0.193***

0.204**

(-1.14)

(3.31)

(2.78)

(-1.19)

(2.98)

(2.31)

D_marriage

-0.058**

-0.087**

0.089*

-0.058**

-0.092**

0.104**

(-2.01)

(-2.04)

(1.89)

(-1.98)

(-2.13)

(2.12)

D_jobtype

0.316***

0.415***

0.291***

0.314***

0.429***

0.284***

(14.98)

(15.20)

(8.28)

(14.83)

(15.59)

(7.65)

_cons

5.118***

4.952***

3.956***

3.836***

4.652***

-1.412

(15.71)

(10.74)

(7.13)

(5.96)

(3.86)

(-0.46)

N

6121

3639

2463

6121

3639

2463

adj. R2

0.232

0.262

0.268

0.226

0.246

0.220

Note: t statistics in parentheses * p < 0.1, ** p < 0.05, *** p < 0.01.

3.4. The Effect of Excess Weight on Income in different sectors

In addition to economic development, occupation may also have differentiated impact on the effect of body size on income. Table 9 presents an inverse u-shaped relationship between BMI and income in both the secondary and tertiary sectors using OLS regressions, with the optimum BMI being much higher in the secondary sector than in the tertiary sector. This may be due to more manual labor in the secondary sector such as mining, manufacturing and construction. The tertiary sector loses significance using IV regressions.

Table 9. Wage regression for different industries.

(1)

(2)

(3)

(4)

(5)

(6)

ols_primaryindustry

ols_secondindustry

ols_tertiary

IV_primaryindustry

IV_secondindustry

IV_tertiary

bmi

-0.008

0.076***

0.042**

0.163

0.101**

0.185

(-0.04)

(3.05)

(2.52)

(0.09)

(2.11)

(1.45)

bmi2

0.001

-0.001**

-0.001*

-0.001

-0.002**

-0.004

(0.25)

(-2.31)

(-1.83)

(-0.04)

(-2.47)

(-1.40)

age

-0.068

-0.020**

-0.009

-0.124

-0.018*

-0.011

(-0.94)

(-2.04)

(-1.16)

(-1.40)

(-1.78)

(-1.36)

age2

0.001

0.000*

-0.000

0.001

0.000

0.000

(1.14)

(1.69)

(-0.11)

(1.43)

(1.53)

(0.17)

ln_edu

0.594*

0.542***

0.579***

0.583**

0.542***

0.576***

(1.95)

(9.00)

(13.16)

(2.10)

(8.95)

(13.12)

ln_occuyear

0.006

0.074***

0.203***

0.019

0.072***

0.202***

(0.08)

(4.81)

(22.82)

(0.24)

(4.72)

(22.65)

health

-0.033

0.018

-0.009

-0.063

0.018

-0.007

(-0.38)

(1.08)

(-0.88)

(-0.62)

(1.12)

(-0.67)

D_year

0.289**

0.115***

0.090***

0.241**

0.113***

0.090***

(2.49)

(5.16)

(7.12)

(2.00)

(5.05)

(6.99)

D_hukou

0.176

-0.273***

-0.076**

0.102

-0.271***

-0.074**

(0.56)

(-4.51)

(-2.26)

(0.26)

(-4.44)

(-2.18)

D_nation

0.964***

-0.114

0.228***

0.885**

-0.118

0.228***

(3.11)

(-0.85)

(3.57)

(2.12)

(-0.89)

(3.58)

D_marriage

-0.281

-0.077*

0.004

-0.465

-0.077*

0.005

(-0.87)

(-1.66)

(0.16)

(-1.38)

(-1.65)

(0.18)

D_jobtype

0.297

0.321***

0.382***

0.305

0.325***

0.381***

(1.34)

(9.06)

(20.89)

(1.41)

(9.16)

(20.56)

_cons

5.607**

5.395***

4.847***

4.546

5.208***

3.293**

(2.44)

(13.00)

(16.63)

(0.22)

(7.79)

(2.30)

N

118

3285

8820

118

3285

8820

adj. R2

0.188

0.129

0.257

0.123

0.123

0.249

Note: t statistics in parentheses * p < 0.1, ** p < 0.05, *** p < 0.01.

Thus, we have provided some discussion of the differential impact of region and occupation on the effect of body size on earnings.

[line 392-408]

“We also find that regions (at different economic stages) and occupations have differentiated impact on the effect of body size on earnings. The effect of being overweight on income is most pronounced in the most developed region of China, and more pronounced in the least developed regions as well. In addition, the optimum BMI is significantly higher in the secondary sector than in the other sectors. Brunello and D’Hombres [64] and Lundborg et al. [42] found that the negative effect of excess weight on labor market outcome is more pronounced in relatively underdeveloped Southern and Central Europe than in relatively developed Northern Europe. The current literature points to differences in dietary intake as the main cause of the gender disparities in overweight in developed countries, and differences in reduced physical activity during the shift from agricultural to wage labor as the main cause of the gender disparities in overweight in developing countries. As for China, while the direction and causes of the weight-wage relationship are similar in the most and least developed regions, the underlying culture is very different. In economically developed and commercially active regions, business banquet out of office is key to negotiating business deals, while in least economically developed regions, excessive dining and drinking at official banquets are crucial to establishing power hierarchies or forming alliances. These are particularly pronounced in China’s highly competitive secondary sector, where “guanxi” (relationships) are needed to secure business partnerships.”

Reviewer 2 Report

The authors used an instrumental variable approach based on a sample of 13,574 individuals from nine provinces in the Chinese Household Income Project (CHIP).

Note that all acronyms/abbreviations must be defined the first time they appear in the abstract, main text, and in tables, for example: BMI.

This is a very interesting study with practical applications and proper statistical analysis. Overall, this is a clear, concise, and well-written manuscript. Several specific comments are as follows.
Literature Review:
For the quality of the literature review, please provide a context for why based on the global obesity pandemic and  the differential impact of different economic stages.

Authors may submit a summary table with the main research papers related to the objective of the manuscript.
What are the research hypotheses? I can't find references on what exactly they are. They should clarify this aspect and focus the analysis of the results towards their contrast.

Under the "Materials & Methods" section, you are required to answer the following:
- What is the response rate?
- What is the sampling technique?

Discussion:
Besides the descriptions of the results of the Hypotheses, more discussion should be offered in more depth.

Conclusions:
Based on these findings, have the current data (yours and others) supported or differentiated? Need to link the results to extant literature, clearly articulate the knowledge gained as results of this study and how this knowledge can be used.

Author Response

Response to Reviewer 2 Comments

Thank you for your careful evaluation of the manuscript. We sent the revised manuscript according to suggestions and answers to the questions below.

Point 1: The authors used an instrumental variable approach based on a sample of 13,574 individuals from nine provinces in the Chinese Household Income Project (CHIP).

Note that all acronyms/abbreviations must be defined the first time they appear in the abstract, main text, and in tables, for example: BMI.

Response 1: Thanks for spotting this. We have checked the manuscript and defined all acronyms/abbreviations as they first appear.

Point 2: This is a very interesting study with practical applications and proper statistical analysis. Overall, this is a clear, concise, and well-written manuscript.

Several specific comments are as follows.

Literature Review: For the quality of the literature review, please provide a context for why based on the global obesity pandemic and the differential impact of different economic stages.

Response 2: Thanks for the comment. We note from the literature (as evident in the summary table of main articles) that countries with higher incomes and higher female labour force participation rates tend to find that women suffer from more pronounced weight wage penalties than men. Therefore, we developed hypothesis 3 that the effect of excess weight on income is related to stages of economic development. We have added relevant context in the introductory section.

[line 38-46]

“Global survey data show that the overall prevalence of obesity is higher among women than men, yet in high-income developed countries, the prevalence of overweight is higher among men than women [3-5]. Current knowledge suggests that different socioeconomic dynamics around the world contribute to the complexity of gender disparities in excess weight gain. In developed countries, where most occupational roles have been sedentary for the past half century with little gender difference, dietary intake rather than physical activities is responsible for gender disparities in overweight and obesity. However, in developing countries, the transition from agricultural labour to wage labour has led to a greater reduction in women’s physical activity than men’s, and the reduction has been particularly pronounced in rural areas [4].”

[line 99-104]

“Along with an increased intake of sugar-laden foods, women, often more physically inactive than men, appear to be more susceptible to the effects of these foods on obesity. Yet, countries with higher incomes and higher female labour force participation rates tend to find that female obesity rates rise more slowly than those of men [4], and that women suffer more pronounced weight penalties in terms of wages than men.”

[line 116]

Hypothesis 3. The effect of excess weight on income is related to stages of economic development.”

[line 127]

“(3) The effect of excess weight on income is most pronounced in Eastern China, which is the most economically developed and commercially active region in China.”

Point 3: Authors may submit a summary table with the main research papers related to the objective of the manuscript.

Response 3: Thanks for the suggestion. We have included a summary table of main research articles related to excess weight and labour market outcomes to our manuscript in section “Introduction”.

[line 96]

Table 1. Main research articles related to excess weight and labor market outcomes.

Reference

Countries

/Regions

Results

Relationship between body size and labor market outcomes

Gender Disparities

Averett and Korenman [32]

United States

There are economic penalties to being overweight, but the penalty is much smaller among black women.

Obese women have lower family incomes than women whose weight-for-height is in the “recommended” range, while results for men are weaker and mixed.

Cawley [34,51]

United States

Both body mass and weight have negative wage effect, but the significance of this effect is not obvious among Hispanic workers.

The negative impact of body mass and weight on wages is largest for white females and smallest for black females.

Larose et al. [62]

Canada

Among working-age adults, obesity led to larger reductions in hourly wages and annual earnings for women than for men.

Brunello and D’Hombres [64]

Pooling 9 European countries

Body mass has a negative effect on earnings, and the impact is larger and statistically more significant in Southern Europe than in Northern Europe.

A 10% increase in the average BMI reduces the real earnings of men and women by 3.27% and 1.86%, respectively.

Lundborg et al. [42]

Pooling 10 European countries

Obesity has significant negative effect on the probability of being employed, and the effect is most pronounced for men in Southern and Central Europe.

Obese European women earned 10% less than their non-obese peers, while for men the effect is smaller and insignificant. Obese women in central Europe faced the greatest wage penalty.

Sargent and Blanchflower [31]

Great Britain

There is a statistically significant inverse relationship between obesity and earnings for women, while there are no obesity effects for men.

Morris [37]

England

BMI has a negative and significant effect on occupational attainment in women, while the results for men are mixed.

Kropfhauber

and Sunder [72]

Germany

There is an inverse u-shaped relationship between BMI and log wages.

The optimum BMI for wage is achieved at 30 for men and 27 for women.

Greve [45]

Denmark

In the private sector, BMI has a negative effect on wages for women but an inverse u-shaped effect on wages for men, whereas results from the public sector show that BMI has no influence on wages for either men or women.

Johansson et al. [46]

Finland

Waist circumference has a negative association with wages for women, whereas no obesity measure is significant in the linear wage models for men.

Obesity is negatively associated with women’s employment probability and fat mass is negatively associated with men’s employment probability.

Dackehag et al. [77]

Sweden

There is a significant obesity penalty in income for men, but no significant excess-weight penalty for women.

Haddad and Bouis [68]

A southern Philippine province

There is a positive relationship between body size and labor productivity as measured by agricultural wages.

N/A

Thomas and Strauss [69]

Brazil

Health measures, such as BMI and per capita calorie intake, positively and significantly affected wages.

BMI affected only men’s wage.

Shimokawa [73]

China

There is a wage penalty for very heavy and thin persons.

The wage penalty is more significant among men than among women.

Pan et al. [74]

China

Body size has an inverse u-shaped effect on the probability of being employed.

The optimal BMI for employment is estimated to be 24.3 for male and 22.7 for female.

Point 4: What are the research hypotheses? I can't find references on what exactly they are. They should clarify this aspect and focus the analysis of the results towards their contrast.

Response 4: Thanks for the comment. We apologize for our lack of clarity.  We have set out the research hypotheses in section “Introduction”, and focused our analysis of the results around them.

[line 111-116]

Here, we put forth three hypotheses:

Hypothesis 1. There is an inverse u-shaped relationship between excess weight and wages, with employment settings being most favorable when the body size is moderate.

Hypothesis 2. Women are more likely than men to be penalized in terms of wages for being overweight or obese.

Hypothesis 3. The effect of excess weight on income is related to stages of economic development.

Point 5: Under the "Materials & Methods" section, you are required to answer the following:

- What is the response rate?

- What is the sampling technique?

Response 5: Thanks for these important suggestions. For this manuscript we use data from the Urban House Survey conducted by China’s National Bureau of Statistics (NBS) as part of the collaborative research project, the Chinese Household Income Project (CHIP), on incomes and inequality in China. We have added its sampling methods to our manuscript in subsection “Data” of section “Materials and Methods”.

[line 152-167]

“The nine provinces were selected with the condition that they should constitute a representative sample of the economic characteristics in urban China’s various regions. The respondent households are selected using a two-stage stratified systematic random sampling scheme. In the first stage, all cities and county towns are categorized by population size, divided into geographical regions, and in each region the cities and county towns of each category are arranged according to the average wages of their staff and workers. The number of individuals who are staff and workers in the cities is summed and the sample cities or counties are selected using an interval of one million staff and workers. In the second stage, a multiphase sampling scheme was used to select sample households within the selected cities and county towns, through sample subdistrict and sample resident committees.

The sample was interviewed through questionnaires designed by the project team. Detailed information was collected on household demographics, educational attainments, employment status, health conditions, social and economic characteristics at both personal and household level. We choose to use data from the 2007 and 2008 surveys because these are the only surveys that include information on body size; other surveys, including the most recent 2013 survey, did not collect information on weight and height.”

For this manuscript, unfortunately, we were not able to find the response rate in the CHIP database or publications based on CHIP data.

Point 6: Discussion: Besides the descriptions of the results of the Hypotheses, more discussion should be offered in more depth.

Response 6: Thanks for the recommendation. We have examined the effect of excess weight on income by region (at different economic stages) and by sector, and included them to our manuscript as section 3.3 and section 3.4. Thus, we have provided some discussion of the differential impact of region and occupation on the effect of body size on earnings.

[line 392-408]

“We also find that regions (at different economic stages) and occupations have differentiated impact on the effect of body size on earnings. The effect of being overweight on income is most pronounced in the most developed region of China, and more pronounced in the least developed regions as well. In addition, the optimum BMI is significantly higher in the secondary sector than in the other sectors. Brunello and D’Hombres [64] and Lundborg et al. [42] found that the negative effect of excess weight on labor market outcome is more pronounced in relatively underdeveloped Southern and Central Europe than in relatively developed Northern Europe. The current literature points to differences in dietary intake as the main cause of the gender disparities in overweight in developed countries, and differences in reduced physical activity during the shift from agricultural to wage labor as the main cause of the gender disparities in overweight in developing countries. As for China, while the direction and causes of the weight-wage relationship are similar in the most and least developed regions, the underlying culture is very different. In economically developed and commercially active regions, business banquet out of office is key to negotiating business deals, while in least economically developed regions, excessive dining and drinking at official banquets are crucial to establishing power hierarchies or forming alliances. These are particularly pronounced in China’s highly competitive secondary sector, where “guanxi” (relationships) are needed to secure business partnerships.”

We have also elaborated a bit more on the sociocultural complexity of the relationship between excess dining and income.

[line 384-390]

“We suspect that the excess-weight premium in wages for Chinese urban men is driven by entrenched business practices of excessive dining and drinking associated with senior positions. Food plays a major role in Chinese culture and is often seen as a form of strengthening intimacy between business associates. Not only are formal business meals, often including twelve to sixteen dishes served with white spirits, a key method for establishing and maintaining strong relationships, but it is not unusual for important business decisions to be made at dinners or banquets outside the office [76,77].”

Point 7: Conclusions: Based on these findings, have the current data (yours and others) supported or differentiated? Need to link the results to extant literature, clearly articulate the knowledge gained as results of this study and how this knowledge can be used.

Response 7: Thanks for the suggestion. We have rewritten the concluding section based on the points you raised.

[line 421-440]

“This article investigates the excess-weight penalty in wages for men and women in China’s labour market. We find a 7% excess-weight premium in wages for overweight men and a 4.6% penalty for overweight women, compared to their healthy-weight peers. We also find an inverse u-shaped association between BMI and logarithm of monthly income for men, with an implied optimum achieved at a BMI of 30, i.e., above the threshold of obesity, while women are better off the slimmer they are. The CHIP survey, initially set up to investigate income and inequality in China, collected information on weight and height for 2007 and 2008, which made possible our study linking body size to wages. Our results are generally consistent with current literature on excess-weight wage penalties in China, including Shimokawa’s finding of a wage penalty for very heavy and thin persons among both men and women but the penalty was more pronounced for men [73], and Pan, Qin and Liu’s finding of an inverse u-shaped effect of body size on the probability of being employed [74]. Our focus, however, is primarily on gender disparities, and the underlying sociocultural complexities. China is on the front lines of the battle against obesity and policies aimed at reducing obesity must be adapted to its unique sociocultural context in order to have gender-differentiated effects. Obesity-related health campaigns alone may exacerbate the already existing discrimination against overweight female, but anti-corruption campaign or, more recently, anti-food-waste campaign may help reduce the weight-wage premium for men, thereby reversing the obesity epidemic and gender disparity. Overall, given the small number of studies to date, the picture of how men are rewarded for being overweight is incomplete. There is clearly a need for further research on occupational attainment and gender disparities in China’s labour market.”

Reviewer 3 Report

Comments on “Wage penalties or wage premiums? A socioeconomic analysis of gender disparity in obesity in urban China”.

The paper presents an interesting topic on “gender disparity in obesity”, using the case of urban China. The overall quality of the paper is high, it is based on innovative methodology and the results are rich but not enough contextualized. The article provides original and new contribution to its theme but needs to be more contextualized.

Despite the promising implications of the paper and an overall sound methodological framework I rise a couple of points that might bring improvements to the paper

  1. Given that is still a topic raising much interest, can the authors add a short table that summarizes some of the key papers reviewed with the main result highlighted in those papers, and presented in this paper? Especially for the long list of references provided, p. 2. (references 27 to 64).
  2. According to the World Health Organization, the indicator for being underweight is defined as a BMI less than 18.5; overweight as a BMI between 25 and 30; and obese as BMI of 30 or higher. So, why authors define the indicator for being obese as a BMI of 28 or higher? It needs to be explained.
  3. The results could be better contextualized. It would be possible to comment on the results obtained by the authors with other studies bringing, for example, geographical distinctions according to Chinese regions. See for example: Linfeng Zhang, Zengwu Wang, Xin Wang, Zuo Chen, Lan Shao, Ye Tian, Congyi Zheng, Suning Li, Manlu Zhu, Runlin Gao, Prevalence of overweight and obesity in China: Results from a cross-sectional study of 441 thousand adults, 2012–2015, Obesity Research & Clinical Practice, 10.1016/j.orcp.2020.02.005, (2020).
  4. One of the main interesting results of this article is that men face significant overweight and obesity premium. To explain this phenomenon, authors “suspect that the excess-weight premium in wages for Chinese urban men is driven by entrenched business practices of excessive dining and drinking associated with senior positions.”. The explanation is a little short and it seems necessary to develop the interpretation of these results. The results of this article would benefit from being put into perspective with more qualitative, even ethnographic work on “business practices of excessive dining and drinking associated with senior positions”.

Author Response

Response to Reviewer 3 Comments

Thank you for your careful evaluation of the manuscript. We sent the revised manuscript according to suggestions and answers to the questions below.

Point 1: The paper presents an interesting topic on “gender disparity in obesity”, using the case of urban China. The overall quality of the paper is high, it is based on innovative methodology and the results are rich but not enough contextualized. The article provides original and new contribution to its theme but needs to be more contextualized.

Despite the promising implications of the paper and an overall sound methodological framework I rise a couple of points that might bring improvements to the paper.

    Given that is still a topic raising much interest, can the authors add a short table that summarizes some of the key papers reviewed with the main result highlighted in those papers, and presented in this paper? Especially for the long list of references provided, p. 2. (references 27 to 64).

Response 1: Thanks for the recommendation. We have included a summary table of main research articles related to excess weight and labour market outcomes to our manuscript in section “Introduction”.

[line 96]

Table 1. Main research articles related to excess weight and labor market outcomes.

Reference

Countries

/Regions

Results

Relationship between body size and labor market outcomes

Gender Disparities

Averett and Korenman [32]

United States

There are economic penalties to being overweight, but the penalty is much smaller among black women.

Obese women have lower family incomes than women whose weight-for-height is in the “recommended” range, while results for men are weaker and mixed.

Cawley [34,51]

United States

Both body mass and weight have negative wage effect, but the significance of this effect is not obvious among Hispanic workers.

The negative impact of body mass and weight on wages is largest for white females and smallest for black females.

Larose et al. [62]

Canada

Among working-age adults, obesity led to larger reductions in hourly wages and annual earnings for women than for men.

Brunello and D’Hombres [64]

Pooling 9 European countries

Body mass has a negative effect on earnings, and the impact is larger and statistically more significant in Southern Europe than in Northern Europe.

A 10% increase in the average BMI reduces the real earnings of men and women by 3.27% and 1.86%, respectively.

Lundborg et al. [42]

Pooling 10 European countries

Obesity has significant negative effect on the probability of being employed, and the effect is most pronounced for men in Southern and Central Europe.

Obese European women earned 10% less than their non-obese peers, while for men the effect is smaller and insignificant. Obese women in central Europe faced the greatest wage penalty.

Sargent and Blanchflower [31]

Great Britain

There is a statistically significant inverse relationship between obesity and earnings for women, while there are no obesity effects for men.

Morris [37]

England

BMI has a negative and significant effect on occupational attainment in women, while the results for men are mixed.

Kropfhauber

and Sunder [72]

Germany

There is an inverse u-shaped relationship between BMI and log wages.

The optimum BMI for wage is achieved at 30 for men and 27 for women.

Greve [45]

Denmark

In the private sector, BMI has a negative effect on wages for women but an inverse u-shaped effect on wages for men, whereas results from the public sector show that BMI has no influence on wages for either men or women.

Johansson et al. [46]

Finland

Waist circumference has a negative association with wages for women, whereas no obesity measure is significant in the linear wage models for men.

Obesity is negatively associated with women’s employment probability and fat mass is negatively associated with men’s employment probability.

Dackehag et al. [77]

Sweden

There is a significant obesity penalty in income for men, but no significant excess-weight penalty for women.

Haddad and Bouis [68]

A southern Philippine province

There is a positive relationship between body size and labor productivity as measured by agricultural wages.

N/A

Thomas and Strauss [69]

Brazil

Health measures, such as BMI and per capita calorie intake, positively and significantly affected wages.

BMI affected only men’s wage.

Shimokawa [73]

China

There is a wage penalty for very heavy and thin persons.

The wage penalty is more significant among men than among women.

Pan et al. [74]

China

Body size has an inverse u-shaped effect on the probability of being employed.

The optimal BMI for employment is estimated to be 24.3 for male and 22.7 for female.

Point 2: According to the World Health Organization, the indicator for being underweight is defined as a BMI less than 18.5; overweight as a BMI between 25 and 30; and obese as BMI of 30 or higher. So, why authors define the indicator for being obese as a BMI of 28 or higher? It needs to be explained.

Response 2: Thanks for spotting this. According to the World Health Organization (WHO), overweight and obesity are defined as 25 ≤ BMI < 30 and BMI ≥ 30, respectively. However, there is growing evidence that the WHO cutpoints derived from the white population may not apply to Asian population, particularly the Chinese working-age population. Since our focus in limited to the impact of body size on labour market performance of the working-age population, here we use overweight and obesity from the Working-Group of Obesity in China (WGOC). We have added this as a footnote to line 202.

Point 3: The results could be better contextualized. It would be possible to comment on the results obtained by the authors with other studies bringing, for example, geographical distinctions according to Chinese regions. See for example: Linfeng Zhang, Zengwu Wang, Xin Wang, Zuo Chen, Lan Shao, Ye Tian, Congyi Zheng, Suning Li, Manlu Zhu, Runlin Gao, Prevalence of overweight and obesity in China: Results from a cross-sectional study of 441 thousand adults, 2012–2015, Obesity Research & Clinical Practice,

10.1016/j.orcp.2020.02.005, (2020).

Response 3: Thanks for the suggestion. We have examined the effect of excess weight on income by region (at different economic stages) and by sector, and included them to our manuscript as section 3.3 and section 3.4.

[line 338-357]

3.3. The Effect of Excess Weight on Income in different regions

As an economy in transition, different regions of China are at different stages of economic development. Some eastern provinces such as Shanghai, Guangdong and Zhejiang have GDP per capita at or near the level of developed economies, presenting a completely different economic landscape from that of the central and western regions. Regressions on the effect of excess weight on income in different regions show that excess weight has the greatest effect on income in eastern China. For the least developed western region, the effect of BMI on income is inverse u-shaped but with lower significance, while there is no effect in the central region. Table 8 presents the results of the wage regression for different regions.

Table 8. Wage regression for different regions.

(1)

(2)

(3)

(4)

(5)

(6)

ols_east

ols_middle

ols_west

IV_east

IV_middle

IV_west

bmi

0.075***

0.042*

0.037

0.192***

0.086

0.531*

(3.97)

(1.77)

(0.87)

(3.68)

(0.84)

(1.89)

bmi2

-0.001***

-0.001

-0.000

-0.004***

-0.002

-0.011*

(-3.19)

(-1.33)

(-0.47)

(-3.65)

(-1.01)

(-1.86)

age

-0.009

-0.017

0.013

-0.010

-0.016

0.005

(-1.18)

(-1.31)

(1.22)

(-1.30)

(-1.12)

(0.39)

age2

-0.000

0.000

-0.000

-0.000

0.000

-0.000

(-0.24)

(0.74)

(-1.59)

(-0.15)

(0.71)

(-0.82)

ln_edu

0.588***

0.493***

0.706***

0.582***

0.491***

0.715***

(11.36)

(9.05)

(8.44)

(11.33)

(9.02)

(8.60)

ln_occuyear

0.190***

0.177***

0.147***

0.190***

0.175***

0.149***

(18.95)

(12.41)

(9.94)

(18.92)

(12.03)

(9.75)

health

-0.037***

-0.022

-0.044***

-0.035***

-0.021

-0.040**

(-3.31)

(-1.48)

(-2.76)

(-3.10)

(-1.41)

(-2.44)

D_year

0.094***

0.083***

0.158***

0.091***

0.081***

0.161***

(6.26)

(4.55)

(7.33)

(6.04)

(4.41)

(7.22)

D_hukou

-0.045

0.009

0.020

-0.039

0.012

0.022

(-1.33)

(0.14)

(0.29)

(-1.16)

(0.18)

(0.30)

D_nation

-0.109

0.212***

0.221***

-0.112

0.193***

0.204**

(-1.14)

(3.31)

(2.78)

(-1.19)

(2.98)

(2.31)

D_marriage

-0.058**

-0.087**

0.089*

-0.058**

-0.092**

0.104**

(-2.01)

(-2.04)

(1.89)

(-1.98)

(-2.13)

(2.12)

D_jobtype

0.316***

0.415***

0.291***

0.314***

0.429***

0.284***

(14.98)

(15.20)

(8.28)

(14.83)

(15.59)

(7.65)

_cons

5.118***

4.952***

3.956***

3.836***

4.652***

-1.412

(15.71)

(10.74)

(7.13)

(5.96)

(3.86)

(-0.46)

N

6121

3639

2463

6121

3639

2463

adj. R2

0.232

0.262

0.268

0.226

0.246

0.220

Note: t statistics in parentheses * p < 0.1, ** p < 0.05, *** p < 0.01.

3.4. The Effect of Excess Weight on Income in different sectors

In addition to economic development, occupation may also have differentiated impact on the effect of body size on income. Table 9 presents an inverse u-shaped relationship between BMI and income in both the secondary and tertiary sectors using OLS regressions, with the optimum BMI being much higher in the secondary sector than in the tertiary sector. This may be due to more manual labor in the secondary sector such as mining, manufacturing and construction. The tertiary sector loses significance using IV regressions.

Table 9. Wage regression for different industries.

(1)

(2)

(3)

(4)

(5)

(6)

ols_primaryindustry

ols_secondindustry

ols_tertiary

IV_primaryindustry

IV_secondindustry

IV_tertiary

bmi

-0.008

0.076***

0.042**

0.163

0.101**

0.185

(-0.04)

(3.05)

(2.52)

(0.09)

(2.11)

(1.45)

bmi2

0.001

-0.001**

-0.001*

-0.001

-0.002**

-0.004

(0.25)

(-2.31)

(-1.83)

(-0.04)

(-2.47)

(-1.40)

age

-0.068

-0.020**

-0.009

-0.124

-0.018*

-0.011

(-0.94)

(-2.04)

(-1.16)

(-1.40)

(-1.78)

(-1.36)

age2

0.001

0.000*

-0.000

0.001

0.000

0.000

(1.14)

(1.69)

(-0.11)

(1.43)

(1.53)

(0.17)

ln_edu

0.594*

0.542***

0.579***

0.583**

0.542***

0.576***

(1.95)

(9.00)

(13.16)

(2.10)

(8.95)

(13.12)

ln_occuyear

0.006

0.074***

0.203***

0.019

0.072***

0.202***

(0.08)

(4.81)

(22.82)

(0.24)

(4.72)

(22.65)

health

-0.033

0.018

-0.009

-0.063

0.018

-0.007

(-0.38)

(1.08)

(-0.88)

(-0.62)

(1.12)

(-0.67)

D_year

0.289**

0.115***

0.090***

0.241**

0.113***

0.090***

(2.49)

(5.16)

(7.12)

(2.00)

(5.05)

(6.99)

D_hukou

0.176

-0.273***

-0.076**

0.102

-0.271***

-0.074**

(0.56)

(-4.51)

(-2.26)

(0.26)

(-4.44)

(-2.18)

D_nation

0.964***

-0.114

0.228***

0.885**

-0.118

0.228***

(3.11)

(-0.85)

(3.57)

(2.12)

(-0.89)

(3.58)

D_marriage

-0.281

-0.077*

0.004

-0.465

-0.077*

0.005

(-0.87)

(-1.66)

(0.16)

(-1.38)

(-1.65)

(0.18)

D_jobtype

0.297

0.321***

0.382***

0.305

0.325***

0.381***

(1.34)

(9.06)

(20.89)

(1.41)

(9.16)

(20.56)

_cons

5.607**

5.395***

4.847***

4.546

5.208***

3.293**

(2.44)

(13.00)

(16.63)

(0.22)

(7.79)

(2.30)

N

118

3285

8820

118

3285

8820

adj. R2

0.188

0.129

0.257

0.123

0.123

0.249

Note: t statistics in parentheses * p < 0.1, ** p < 0.05, *** p < 0.01.

Thus, we have provided some discussion of the differential impact of region and occupation on the effect of body size on earnings.

[line 392-408]

“We also find that regions (at different economic stages) and occupations have differentiated impact on the effect of body size on earnings. The effect of being overweight on income is most pronounced in the most developed region of China, and more pronounced in the least developed regions as well. In addition, the optimum BMI is significantly higher in the secondary sector than in the other sectors. Brunello and D’Hombres [64] and Lundborg et al. [42] found that the negative effect of excess weight on labor market outcome is more pronounced in relatively underdeveloped Southern and Central Europe than in relatively developed Northern Europe. The current literature points to differences in dietary intake as the main cause of the gender disparities in overweight in developed countries, and differences in reduced physical activity during the shift from agricultural to wage labor as the main cause of the gender disparities in overweight in developing countries. As for China, while the direction and causes of the weight-wage relationship are similar in the most and least developed regions, the underlying culture is very different. In economically developed and commercially active regions, business banquet out of office is key to negotiating business deals, while in least economically developed regions, excessive dining and drinking at official banquets are crucial to establishing power hierarchies or forming alliances. These are particularly pronounced in China’s highly competitive secondary sector, where “guanxi” (relationships) are needed to secure business partnerships.”

Point 4: One of the main interesting results of this article is that men face significant overweight and obesity premium. To explain this phenomenon, authors “suspect that the excess-weight premium in wages for Chinese urban men is driven by entrenched business practices of excessive dining and drinking associated with senior positions.”. The explanation is a little short and it seems necessary to develop the interpretation of these results. The results of this article would benefit from being put into perspective with more qualitative, even ethnographic work on “business practices of excessive dining and drinking associated with senior positions”.

Response 4: Thanks for the comment. We have elaborated a bit more on the sociocultural complexity of the relationship between excess dining and income in the discussion section.

[line 384-390]

We suspect that the excess-weight premium in wages for Chinese urban men is driven by entrenched business practices of excessive dining and drinking associated with senior positions. Food plays a major role in Chinese culture and is often seen as a form of strengthening intimacy between business associates. Not only are formal business meals, often including twelve to sixteen dishes served with white spirits, a key method for establishing and maintaining strong relationships, but it is not unusual for important business decisions to be made at dinners or banquets outside the office [76,77].”

[line 403-408]

“In economically developed and commercially active regions, business banquet out of office is key to negotiating business deals, while in least economically developed regions, excessive dining and drinking at official banquets are crucial to establishing power hierarchies or forming alliances. These are particularly pronounced in China’s highly competitive secondary sector, where “guanxi” (relationships) are needed to secure business partnerships.”

Given the small number of studies to date, the picture of how men are rewarded for being overweight is incomplete. There is a need for further research on occupational attainment and gender disparities in China’s labor market.

Round 2

Reviewer 1 Report

Comments for the 2nd review:

Some of the recommendations of the previous review have not been revised. It is suggested that the authors must follow the suggestions and recommendations to modify the paper in order to be considered acceptance. The specific recommendations are as follows:

  1. The verified data is too old to reflect the current reality in China! In the past decade, China's economy of the high grow has attracted global attention, this study is based on data for 2007 and 2008, because GDP has changed greatly in recent years, so it is recommended that the research data should increase the latest validation data for 2017 and 2018 at least. Compare with the verification data of 2007 and 2008 to provide the contribution and reference value of the article.

  1. The problem has not been solved. The possible problems (proof) of multiple regression models established in table 4 and table 5 should be excluded before it is recommended to use the variance inflation factor to a test of the multicollinearity. Multiple regression models established in table 4 and table 5 should first be excluded from problem (proof), so it is recommended to use the variance inflation factor to a test of the multicollinearity. For example, In Table 3, the Female’s IV model estimated parameter is not a significant. However, in the case of the BMI estimation parameters of the model (4) in Table 5 becoming significant, this may be the result of spurious regression.

Reviewer 2 Report

ok

Author Response

Thank you for your positive comments.